# Ultrasonic Bonding of Multi-Layered Foil Using a Cylindrical Surface Tool

**Keisuke Arimoto [1], Tomohiro Sasaki [1,*], Yuhei Doi [2] and Taewon Kim [2]**

[1]   Graduate School of Science and Technology, Niigata University, Niigata 9502181, Japan;
      kskarmt0410@gmail.com
[2]   Nissan Motor Co., Ltd, Kanagawa 2520012, Japan; y-doi@mail.nissan.co.jp (Y.D.);
      t-kimu@mail.nissan.co.jp (T.K.)
[*]   Correspondence: tomodx@eng.niigata-u.ac.jp; Tel.: +81-262-6710

**Abstract:** A cylindrical tool was applied for ultrasonic bonding of multi-layered copper foil and a copper sheet to prevent damage to the foil during bonding. The strength of the joints bonded with the cylindrical tool was comparable to that of the joints bonded with a conventional knurled tool. The effect of the cylindrical surface tool on bondability was investigated thorough relative motion behaviors between the tool surface and the bonding materials, as well as on bond microstructure evolution. The relative motion was visualized with in-situ observation using a high-speed camera and digital image correlation. At shorter bonding times, relative motions occurred at the bonding interfaces of the foil and the copper sheet. Thereafter, the relative motion between the tool and the bonding material became predominant owing to bond formation at the bonding interface, resulting in a macroscopic plastic flow in the bonded region. This relative motion damaged the foil in knurled tool bonding, and the cylindrical tool achieved bonding without any damage.

**Keywords:** ultrasonic bonding; lithium-ion battery; multi-layered foil; tool geometry; relative motion; plastic flow

## 1. Introduction

Ultrasonic bonding is used for manufacturing electrical products in the automotive industry, such as lithium-ion batteries and wire harnesses, because of the short bonding time associated with the process [1]. The lithium-ion batteries used in electric vehicles consist of a number of stacked battery cells with multi-layered electrodes made of copper or aluminum foil, and the electrodes are bonded together to an outer terminal by means of ultrasonic bonding. Since several hundred battery cells are installed in an electric vehicle to obtain sufficient power, quality control of ultrasonic bonding is an important technical issue. The principle of ultrasonic bonding is that the interfacial friction between the bonding materials and a normal force causes the removal of oxides and metallurgical adhesion. Vibration energy is applied through a bonding tool attached to the tip of a sonotrode, and this vibration induces relative motion between the bonding materials. A number of studies in the literature have focused on the influence of process parameters, including normal force, vibration amplitude and bonding time, on the bondability of various types of metal sheets [2–8]. In the ultrasonic bonding process, micro-bonds are initially formed at the bonding interface owing to the relative motion of the bonding material. The micro-bonds develop owing to shear deformation in the latter bonding stage, resulting in a macroscopic plastic flow in the bonded region. In addition, a few studies have investigated the ultrasonic bonding of lithium-ion battery materials, such as copper sheet and aluminum sheet [9,10] and copper sheet and Ni-plated Cu sheet [11–13]. It has been revealed the initial micro-bonds are formed through the broken Ni layer; thereafter, plastic flow expands the bonded

region. Moreover, the ultrasonic bonding of multi-layered sheets of battery materials has been studied by some researchers [14–16]. These studies have indicated that sufficient metallurgical adhesion and plastic flow can occur at each interface between the sheets. The aforementioned interfacial phenomena are closely related to the relative motions of the bonding materials; moreover, the relative motion between the tool and the bonding material in contact with the tool surface is an important factor in ultrasonic bonding. A few researchers have observed the relative motion behavior by means of image correlation [17] and photonic Doppler velocimetry [18]. These studies have revealed that the interface at which relative motion occurs between materials changes to the interface between the tool tip and the materials as the bonding time increases, and the relative motion between the tool tip and the bonding material significantly influences bonding quality. The surface of the bonding tool is conventionally machined in a knurled pattern to clamp the bonding material and to cause interfacial friction. However, penetration of the knurled edge during the bonding process occasionally damages the bonding materials [19]. In the manufacture of lithium-ion batteries, a cover sheet is generally lapped over to prevent any damage to the underlying multi-layered foil. A few studies on the effects of tool geometry have been reported in the literature. Lee et al. [20] performed finite-element simulations to study the effect of knurl edge geometry, and they showed that large plastic strain is concentrated at the contact point with the knurled edge; such large strain concentration damages the bonding materials. In addition, Komiyama and Sasaki et al. [21,22] investigated the effect of the geometry of the knurled edge on the ultrasonic bonding of aluminum sheet and developed a new knurled edge tool that can achieve higher joint strength with smaller penetration than the conventional knurled edge tool. Watanabe et al. [23] developed a cylindrical surface tool without a knurled edge, and they found out that this tool can prevent damage to the upper bonding metal, resulting in increased joint strength of A6061 aluminum alloy sheet joints. Latter work by our group [24] showed that the relative motion of the bonding tool and the bonding material predominantly influences the bond formation, particularly in case of this cylindrical surface tool. However, the results reported so far on the effect of tool geometry have been obtained from bonding metal sheets thicker than 0.5 mm. In this study, the cylindrical surface tool is applied to the bonding of multi-layered foil to suppress damage to the foil. The objective of the present study is to investigate the effect of the tool geometry on the deformation of multi-layered foil and to reveal the bonding mechanism of such foil. Bonding with the cylindrical surface tool was compared with bonding with the knurled surface tool based on the relative motion between the tools and the bonding materials, as well as microstructure evolution of the bonding materials.

## 2. Experimental Procedure

### 2.1. Specimen

In this study, we used 7.0 μm thick copper foil and a 0.2 mm thick Ni-plated copper sheet (lower sheet). Ni layer with 2.8 μm thick was plated on both side of the bonding surface and the lower surface in contact with the anvil. These materials were fabricated into bonding specimens, as shown in Figure 1. Seventeen pieces of Cu foil (multi-layered foil) were lapped on the lower sheet. For the observation of relative motion, a 0.3 mm thick copper sheet was used instead of the multi-layered foil because it was difficult to analyze each foil with the following image correlation.

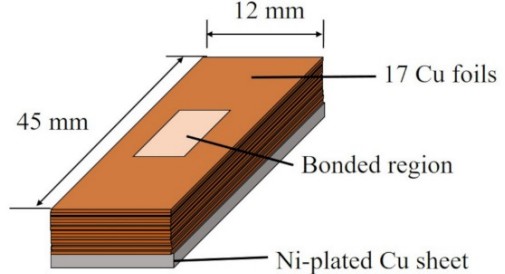

**Figure 1.** Specimen used in this study.

### 2.2. Bonding Test

An ultrasonic bonder with a rated power of 1200 W and vibration frequency of 21 kHz (USW1221G3X1, Ultrasonic engineering Co., Ltd, Tokyo, Japan) was used in the bonding test. Figure 2a shows a schematic illustration of the ultrasonic bonding test. Two ultrasonic sonotrodes with a knurled pattern (K-type) and a cylindrical surface without the knurled edge (C-type) were prepared. The two horns were machined such that the resonant frequencies become 21 kHz ± 0.1 Hz. A schematic illustration of ultrasonic bonding and the tool geometries are shown in Figure 2b–d. The K-type tool shown in Figure 2b has a knurled pattern. In case of the C-type tool, a cylindrical surface with a radius of 200 mm was machined along a direction perpendicular to the vibration direction (direction *y*), as shown in Figure 2c. The amplitudes in the vibration direction of the tool under no load were approximately 29.0 μm in case of the K-type tool and approximately 30.0 μm in case of the C-type tool. The anvil tool tip had a knurled surface as well, as shown in Figure 2d. The bonding test was performed by controlling bonding time under a constant normal load of 588 N. The bonding time ranged from 200 to 500 ms. The bonded specimen was cut at the bonded region, and its cross section was polished. The microstructure of the specimen was observed from two directions (Figure 3). Joint strength was measured using a 'T-shaped' specimen, as shown in Figure 4. Both ends of the specimen were clamped to a tensile machine, and the maximum load in the tensile test was measured. The tensile test was conducted four times for each bonding condition.

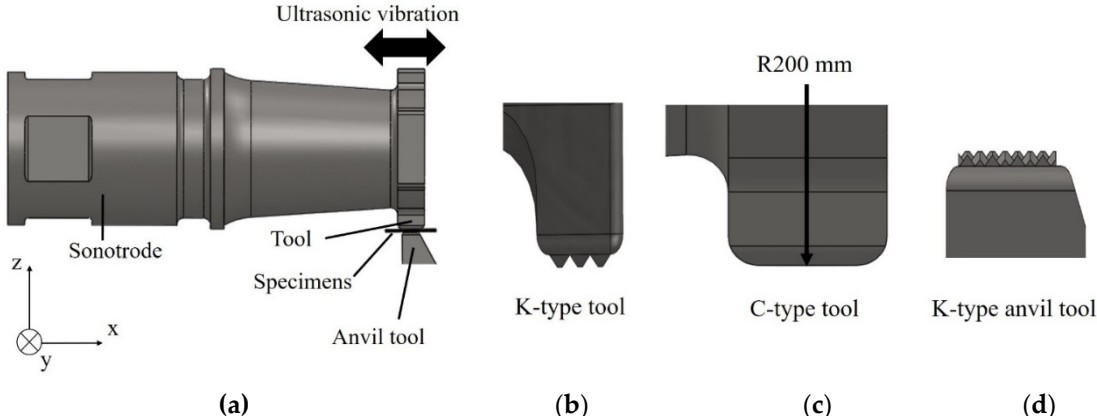

**Figure 2.** Schematic illustration of (**a**) ultrasonic bonding and tool geometries of the (**b**) knurled tool (K-type), (**c**) cylindrical surface tool (C-type) and (**d**) knurled edge anvil tool.

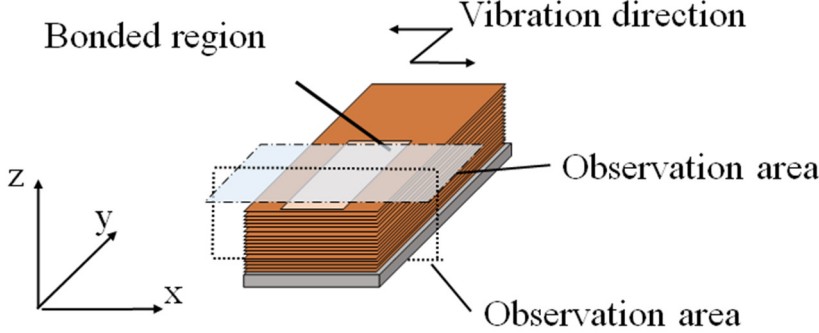

**Figure 3.** Cross-sectional observation area.

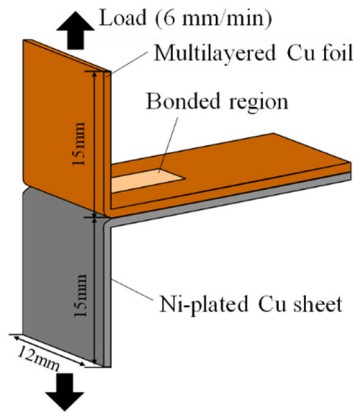

**Figure 4.** Tensile shear test specimen.

*2.3. Relative Motion Analysis*

The behaviors of the tools and the specimens during the ultrasonic bonding test were observed using a high-speed camera (Photron Ltd., Tokyo, Japan). The 0.3 mm thick sheet and the lower sheet were used as bonding materials. The capture rate was 100,000 frames per second. Displacement behaviors were calculated by image correlation [17] from the captured video. Four subset areas of $10 \times 15$ pixels were set for the tool, upper sheet (0.3 mm thick Cu sheet), lower sheet and anvil (Figure 5) to obtain their coordinates. Image intensity was interpolated using a spline curve. The resolution of the interpolated image corresponded to approximately 1.81 μm/pixel. Displacement waveforms in the vibration direction were obtained by conducting spline interpolation. The number of displacement data increased 10-fold upon interpolation. In addition, the amplitude per unit time was calculated through Fourier transformation by extracting displacement waveforms at intervals of 1 ms.

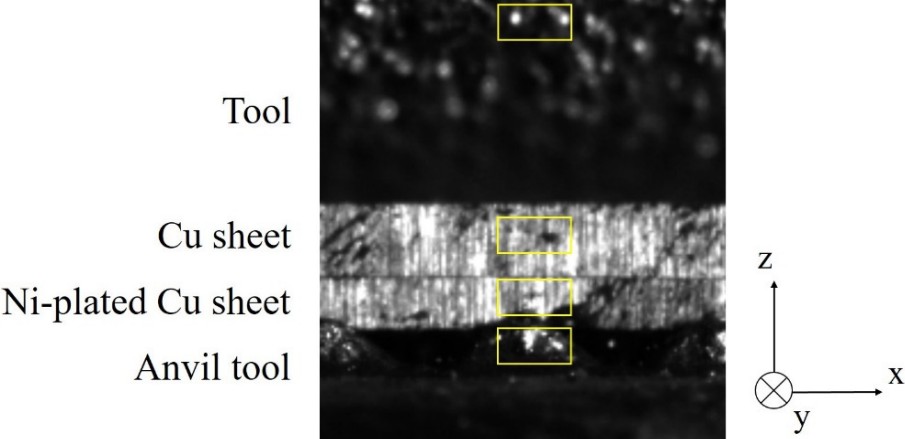

**Figure 5.** Observation area for image correlation.

## 3. Results and Discussions

*3.1. Joint Strength and Damage to Multi-Layered Foil*

Figure 6 shows the joint strength of the bonded specimens plotted against bonding time. For bonding times smaller than 200 ms, the joint strength was too low to conduct a tensile shear test on the specimens. The joint strength increased with increasing bonding time for both tool geometries. No significant differences were observed between the K-type tool and the C-type tool, indicating that sufficient joint strength was achieved using the C-type tool. Figures 7 and 8 show the appearances of the bonded specimens and their fracture patterns, respectively. In the specimen bonded with the K-type tool, edge indentation formed on the bonded surface, causing winkles of foils around the bonding are.

The upper foil was fractured at the corners of the bonded region, marked by a red rectangle in Figure 7a. The foil showed a tendency to incur damage on the right side of the bonded region, which was the far side of the tool from the ultrasonic bonder. The vibration of the sonotrodes involves bending motion, resulting in non-uniform normal stress in the bonding area. Thus, the trend of damage at the far side of the sonotrode may be ascribed to the deflection of the sonotrode. The damage expanded along the outer edge of the tool during bonding times ranging from 300 to 500 ms (Figure 7b–d). By contrast, no damage on the bonding surface was observed in the case of bonding with the C-type tool for all bonding durations (Figure 7e–h). Regarding the fracture pattern obtained with the K-type tool, as shown in Figure 8a–d, the joints fractured around the outer side of the bonded region, and the Cu foil remained on the lower sheet. This implies that joint strength is governed by the strength of the multi-layered Cu foil. In the bonding with the C-type tool for 200 ms (Figure 8e), the bonded region tended to be smaller in the upper foil, indicating that the lower foil has greater bond strength. Moreover, the bonded region expanded with increasing bonding time, and the bonded region was formed over the entire contact area with the C-type tool at bonding durations longer than 400 ms (Figure 8e–h). Expansion of the bonded region in case of the C-type tool is explained based on a discussion of relative motion behavior and microstructural evolution in the following sections.

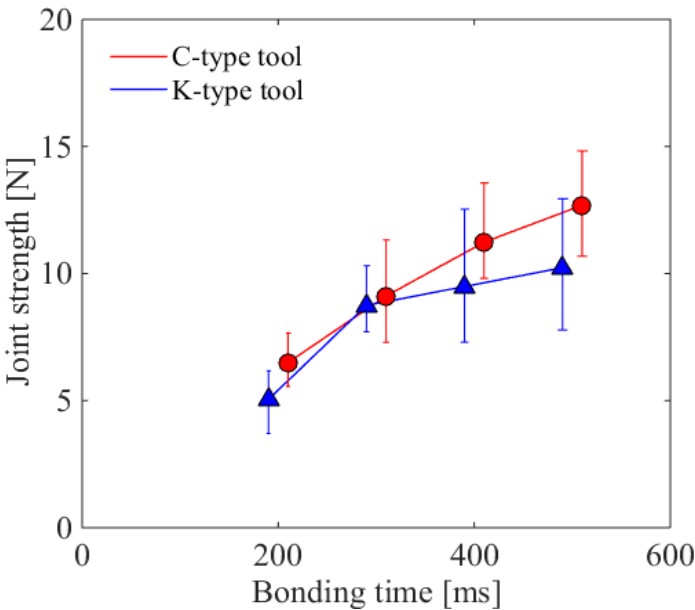

**Figure 6.** Joint strength of multi-layered Cu foil and Ni-plated Cu sheet.

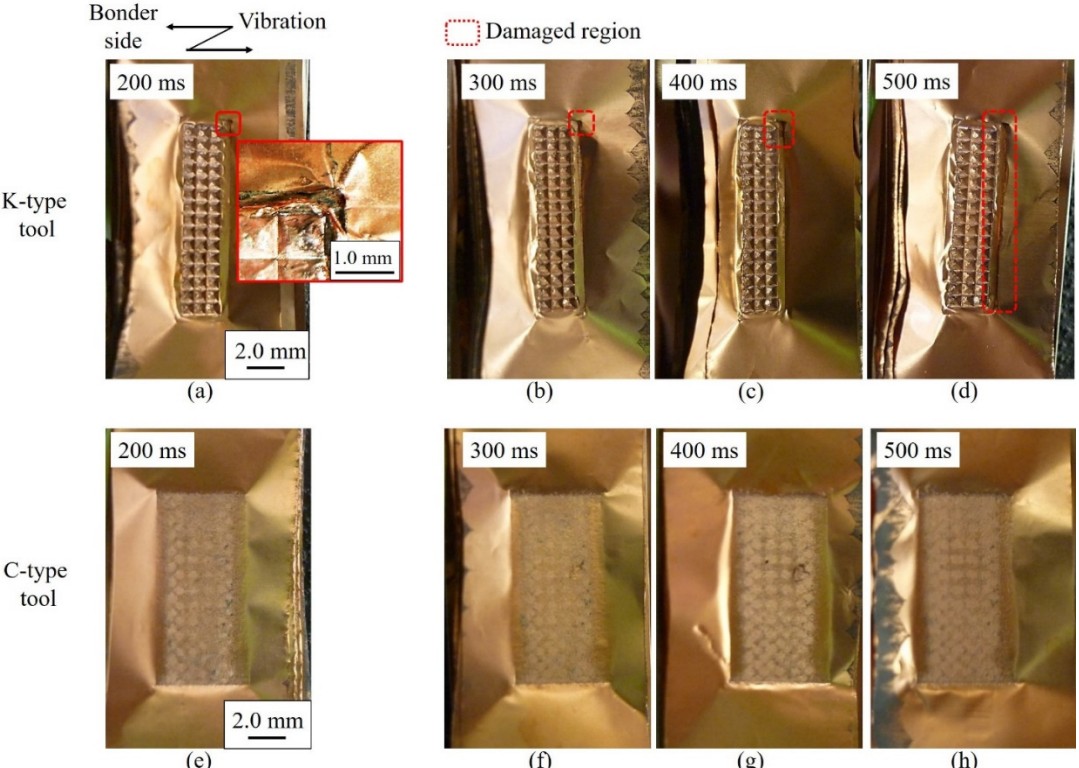

**Figure 7.** Appearance of bonded specimens. (**a**–**d**) and (**e**–**h**) are joints bonded with K-type tool and C-type tool, respectively.

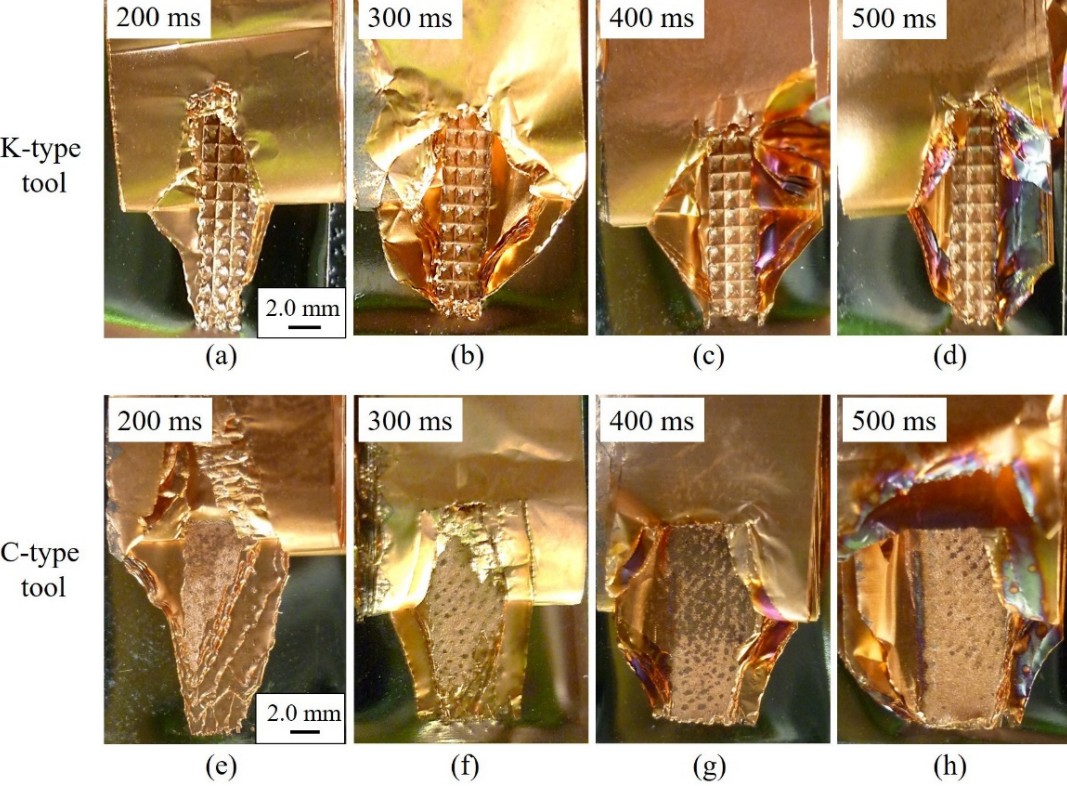

**Figure 8.** Fracture pattern after tensile shear test. (**a**–**d**) and (**e**–**h**) are joints bonded with K-type tool and C-type tool, respectively.

### 3.2. Relative Motion Behaviors of Tools and Bonding Metals

Figures 9 and 10 show the displacement behaviors in the bonding of the lower sheet and the 0.3 mm thick copper sheet instead of the multi-layered foil. The curves in Figure 9 show the displacements in the normal force direction (direction *z*) of the tools and specimens for a bonding time of 500 ms. In the case of bonding with the K-type tool (Figure 9a), the specimens and the anvil tool were almost stationary in the normal direction. Meanwhile, tool displacement decreased with increasing bonding time; the tool moved in the negative direction. This indicates that the tool edge penetrated into the bonding specimen. By contrast, in the case of the C-type tool, the difference in normal displacement between the tool and the specimen was relatively small; tool displacement in the negative direction was approximately half that in case of the K-type tool. We inferred that the smaller penetration of the C-type tool suppressed fracture to the foil, as well as the winkles around bonding area, shown in Figure 7.

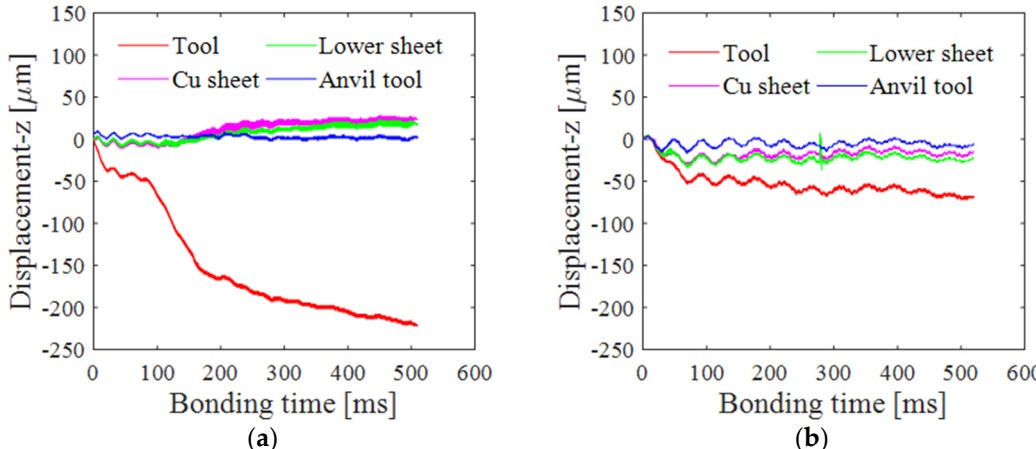

**Figure 9.** Displacement in the normal direction (*z*-axis) (**a**) using the K-type tool and (**b**) using the C-type tool.

Figure 10 shows the displacement curves along the vibration direction for a short bonding time of 30 ms. The figure confirms that the Cu sheet and the tool vibrate with the same amplitude, and the lower sheet and the anvil tool show almost no vibration. This fact is indicative of the occurrence of relative motion at the interface between the Cu sheet and the lower sheet. To assess such movements of tools and specimens during bonding, amplitudes of tools and specimens were calculated from the displacement waveforms for all bonding durations. Figure 11 shows the amplitude variations of the tools and the specimens. The amplitude of the Cu sheet was almost equal to that of the tool until approximately 100 ms in case of the K-type tool and until approximately 50 ms in case of the C-type tool. The relative motion between the Cu sheet and the lower sheet was predominant in this time range. Relative motion with the K-type tool was longer than that with the C-type tool because the edge of the K-type tool penetrated into the Cu sheet and cramped it during initial vibration. After each bonding duration, the amplitude of the Cu sheet decreased to be equal to that of the lower sheet and the anvil tool for each of the tools, indicating the relative motion between the tool and the Cu sheet. This transition of relative motion behavior is consistent with the bonding process of thicker aluminum sheets reported by Sasaki et al. [22,24]. It has also been reported that the relative motion between the bonding tool and the upper specimen generates frictional heat, leading to the formation of the large bonded region. The result in the present study shows that the relative motion between the tool and the Cu sheet became predominant, regardless of tool geometry. Lee et al. [14] confirmed the occurrence of relative motion between the tool and the multi-layered sheet after the stage of relative motion between the four-layer sheets. In the bonding of the multi-layered Cu foil and the lower sheet, it is surmised that the relative motion between the tool and the multi-layered foil becomes predominant

after the earlier bonding stage involving relative motion at each interface between pieces of foil, as well as between the foil and the lower sheet, resulting in the formation of bonded regions. Notably, the bonding strength increased as the bonding time increased, as shown in Figure 6. Comparing the motion behaviors shown in Figure 11 with joint strength, it is concluded that the bonded region was expanded in the stage in which the relative motion between the tool and the Cu sheet became predominant, thus increasing the joint strength.

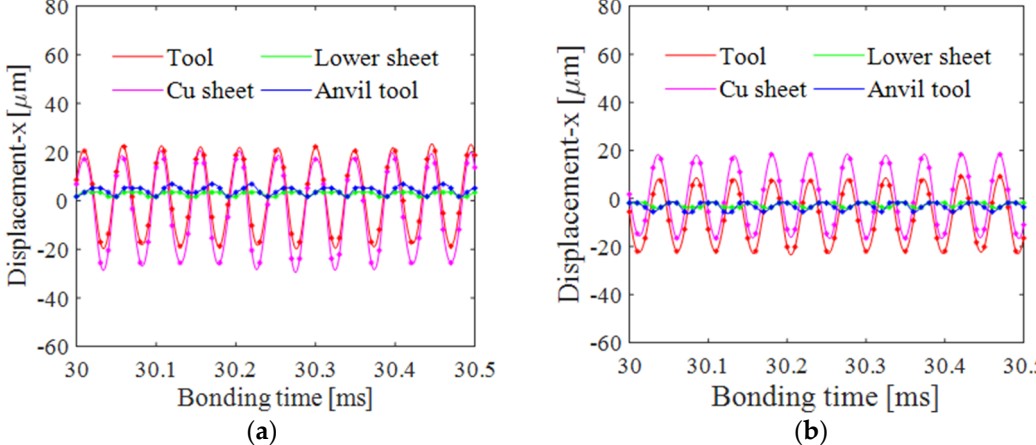

**Figure 10.** Displacement along the vibration direction (*x*-axis) at 30 ms (**a**) using the K-type tool and (**b**) using the C-type tool.

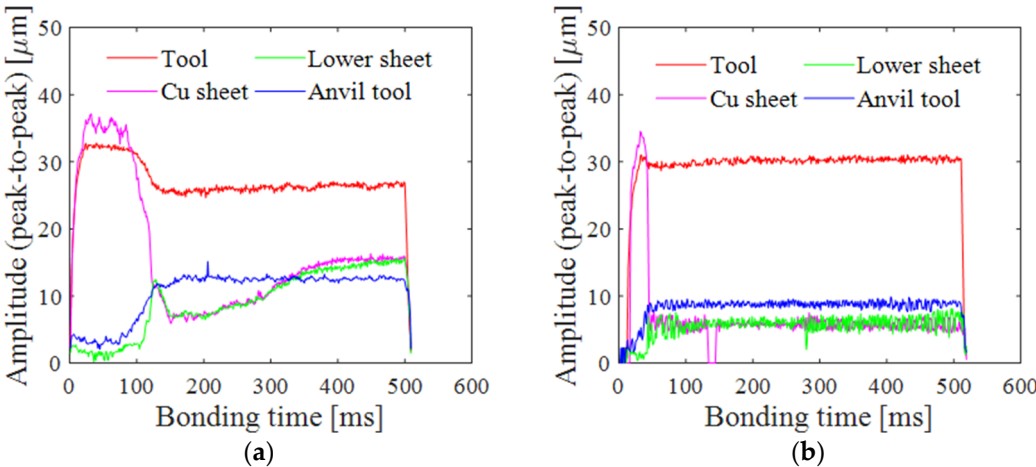

**Figure 11.** Amplitude of tools and specimens when (**a**) using the K-type tool and (**b**) C-type tool.

### 3.3. Evolution of Microstructure

Cross-sectional micrographs of the specimens bonded with the K-type and C-type tools are shown in Figures 12 and 13, respectively. The bonding time was 500 ms. In case of the K-type tool, shown in Figure 12a, severe deformation of the multi-layered foil was observed in the upper part where the tool edge corner came into contact with the foil surface. The edge of the K-type tool penetrated into the upper foil owing to vibration. In addition, as shown Figure 11a, relative motion occurred predominantly between the tool edge and the foil. Therefore, this severe deformation can be ascribed to friction and penetration of the tool edges. Figure 12b shows a magnified version of the image in Figure 12a, as indicated by a white rectangle (A). Notably, the Ni layer under the edge corner was broken. The initial bond may have formed in the earlier bonding stage at the Cu/Cu (foil/foil) and Cu/Ni interfaces by relative motion between each Cu foil, as well as between the Cu foil and the Ni layer. Shear force may occur in the Cu/Ni and Cu/Cu micro-bond regions owing to the relative motion in the

latter bonding stage, resulting in breaking of the Ni layer and formation of Cu/Cu bonds between the foil and the lower sheet. Moreover, a stirred region of the foil and the lower sheet was observed, shown in Figure 12b using a white rectangle (B). Figure 12c shows a magnified version of the aforementioned image around the bonded region. In the stirred region, the boundaries of the Cu foil could still be observed. The anchoring effect contributed to the increase in joint strength by means of plastic flow at the bonding interface. A similar effect on joint strength was reported by Lee et al. [13]. As shown in Figure 11a, the relative motion between the tool and the Cu sheet occurred after 100 ms, leading to the generation of frictional heat. The frictional heat facilitated plastic flow [22]. Therefore, expansion of the bonded regions started after 100 ms owing to plastic flow of the bonding specimens. By contrast, the foil fractured at the point of contact with the corner of the edge, shown in Figure 12b using a white square (C). The relative motion, as described above, caused damage to the Cu foil because the edge of the K-type tool penetrated and came into contact with the Cu foil during ultrasonic bonding, shown using a white square (C). Meanwhile, in case of the C-type tool, damage to the multi-layered foil by the tool edges was not observed, as shown in Figure 13a. In addition, the broken Ni layer and the stirred region can be seen at the top of the anvil tool edge (Figure 13b), implying that the contact between the lower sheet and the anvil tool edge has an influence on the bond formation the surface of C-type tool and the anvil tool edge. This result suggests that stress concentration by the anvil tool edge enhanced the plastic flow due to relative motion between the tool and the surface of multilayered foil.

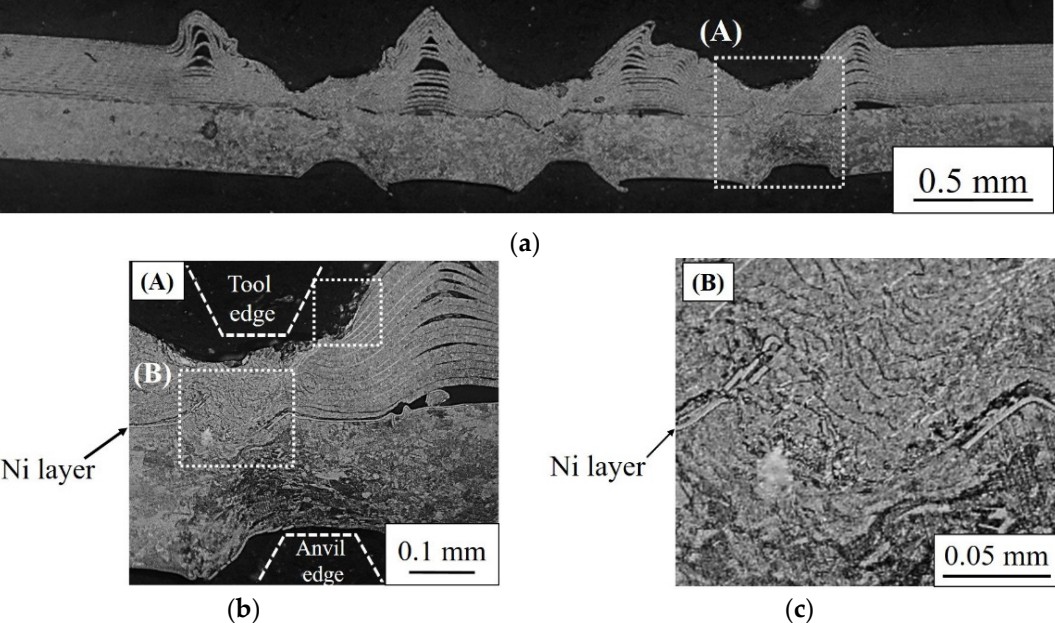

**Figure 12.** Optical micrographs of cross sections when using the K-type tool: (**a**) entire bonded region, (**b**) magnified version of (A) and (**c**) highly magnified version of (B).

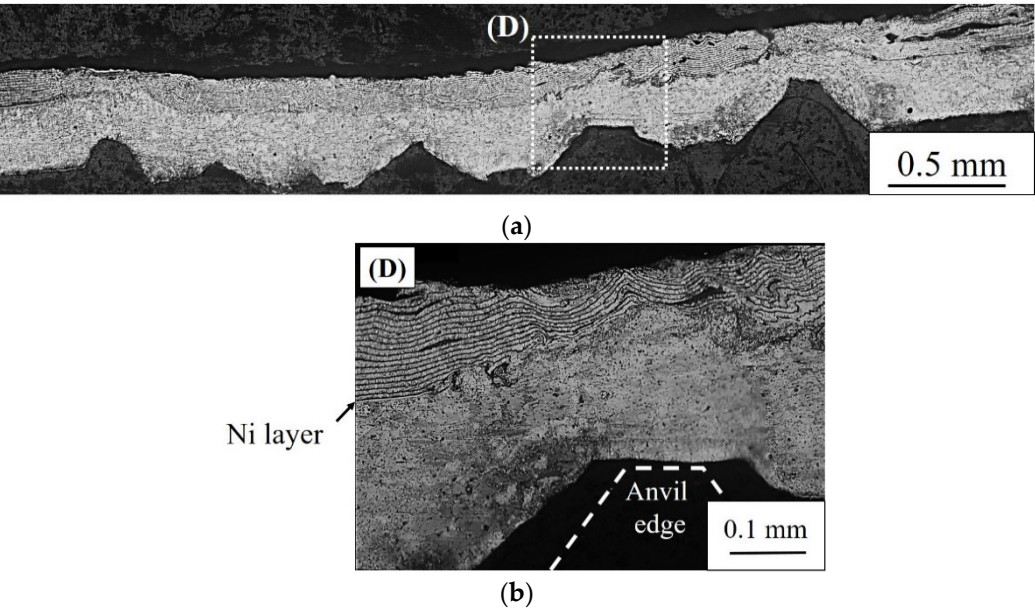

**Figure 13.** Optical micrographs of cross sections obtained using the C-type tool: (**a**) entire bonded region and (**b**) magnified version of (D).

Figure 14 shows cross-sectional micrographs cut parallel to the bonding surface from the Cu foil side. The regions in which the tool edge penetrated are magnified. In case of the K-type tool, the contours of the Cu foil can be observed inside the Ni layer, as denoted by a white circle (Figure 14a); in case of the C-type tool, the Cu foil was outside of the Ni layer (Figure 14b). This difference in the contours of the bonded region was ascribed to the difference in the bond-expansion process. Figure 15 shows a schematic illustration of the bonded region. As shown in Figure 15a, the edge of the K-type tool penetrated into the Cu foil, and the resulting stress pushed them into the lower sheet. In addition, a few foil pieces stirred at the Cu foil side, and other foil pieces were stirred with the lower sheet on the lower sheet side. Therefore, plastic flow occurred from the foil side and, thereafter, from the lower sheet side. By contrast, in the case of bonding with the C-type tool (Figure 15b), the lower sheet was thrust into the Cu foil side by the anvil tool edge. Consequently, expansion of the bonded region occurred by plastic flow of the lower sheet toward the Cu foil side, stirring the Cu foil and the lower sheet; the bonded region expanded from the interface between the foil and the lower sheet to the upper foil. This phenomenon corresponded to the fracture patterns shown in Figure 8e–h.

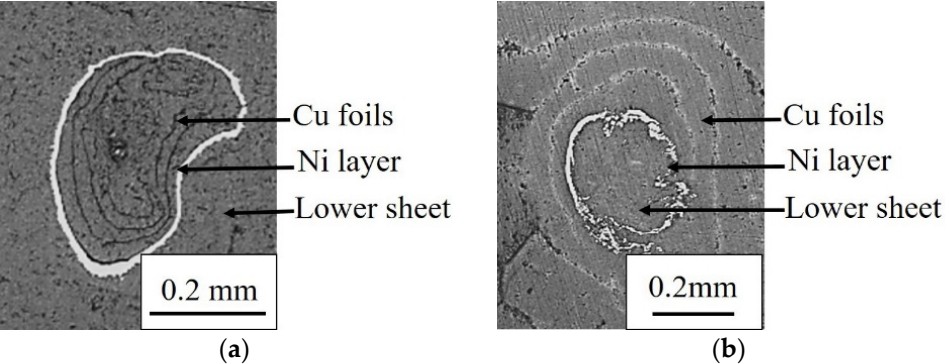

**Figure 14.** Optical micrographs of cross sections from the foil side around the region in which the tool edge penetrated near the bonding interface; (**a**) when using the K-type tool and (**b**) when using the C-type tool.

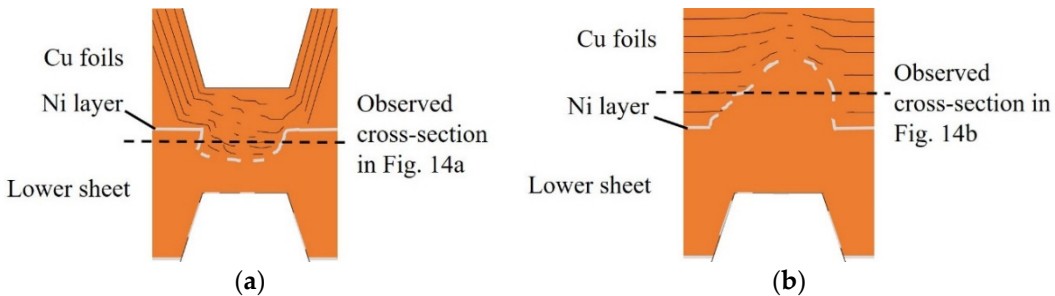

**Figure 15.** Schematic illustration of bonded regions (**a**) using the K-type tool and (**b**) using the C-type tool.

From these results, the bonded region was considered to be formed as follows. First, relative motion between the multi-layered foil and the lower sheet occurred, achieving metallurgical adhesion of each Cu foil and Cu foil/Ni layer. With increasing bonding time, the Ni layer was gradually broken, forming the Cu/Cu bond. Meanwhile, the relative motion at the bonding interface was hindered by bond formation. After this process, the relative motion between the tool and the multi-layered foil became predominant, facilitating plastic flow at the bonding interface. The bonded region expanded with the plastic flow, resulting in increased joint strength. At the same time, stirring of the foil and the lower sheet contributed to an increase in joint strength through the anchoring effect. Regarding the formation of the bonded region, the K-type tool expanded the bonded region from the upper foil, and the C-type tool expanded the bonded region from the interface between the foil and the lower sheet.

## 4. Conclusions

A C-type tool without knurl edges was applied to bonding multi-layered Cu foil, and the effect on deformation of the foil and bond microstructure was investigated from the viewpoint of relative motion behavior during the bonding process. The following conclusions were obtained.

1.  The C-type tool is capable of bonding multi-layered Cu foil and Ni-plated sheet without damaging the upper foil in contact with the tool tip, and the joints made using the C-type tool exhibit adequate joint strength.
2.  The relative motion between the tool and the Cu foil in contact with the tool has a great influence on the bond quality especially in the bonding thin foils, regardless the tool surface geometry. This relative motion facilitates development of the bonded region, while it damages the foil when using the K-type tool. The lower level of penetration of the tool tip in case of the C-type tool led to damage-less bonding.
3.  The micro-bonds formed at the shorter bonding times when the relative motion between pieces of foil, as well as between the foil and the lower sheet, developed along the vibration direction with shear deformation, resulting in macroscopic plastic flow in the bonded region involving the Ni layer for both tool geometries. In case of the K-type tool, plastic flow occurred, especially at the contact point of the edge corner, causing fracture of the foil. Meanwhile, in case of the C-type tool, plastic flow expanded from the lower sheet in contact with the anvil edge.

**Author Contributions:** Conceptualization: T.S.; Investigation: K.A., Y.D., T.K., and T.S.; Project administration: T.S.; Writing, original draft: K.A.; Writing, review and editing: T.S., Y.D., and T.K.

**Conflicts of Interest:** The authors declare no conflict of interest.

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
