# Peer review of "Ultrasonic Bonding of Multi-Layered Foil Using a Cylindrical Surface Tool"

_metals, doi:10.3390/met9050505_

Round 1
Reviewer 1 Report
The application of welding tools with different surface geometry is presented in the submitted paper. The effect of the cylindrical surface tool on bondability was investigated. Authors provided a study of relative motion behaviors between the tool surface and the bonding materials. The investigation of bond microstructure evolution is also presented. The manuscript can be reconsidered for publication after major revision. The following points have to be clarified:
First of all, authors should prove the novelty of their paper. They are requested to emphasize what is original/novel in the submitted manuscript in comparison to literature data and their previous papers. For example, in the Ref. [22] one of the three final conclusions is following:
"In the latter stage of bonding (t > 50 ms) [...] the relative motion between the tool edge and the bonding material becomes predominant."
It is quite similar to second conclusion provided in the submitted paper:
"(2) The relative motion between the tool and the Cu foil became predominant as the bonding time increased." (lines 268-269).
Moreover, the comparison of welding tools with knurled and cylindrical surfaces for ultrasonic bonding was presented in:
Sasaki et al., Effect of tool geometry on ultrasonic welding process, IOP Conf. Series: Materials Science and Engineering 61 (2014) 012006.
What are the working frequencies of used K-type and C-type tools? Authors mentioned that the vibration frequency of an ultrasonic bonder was 21 kHz. However, a resonant frequency of an ultrasonic system can be changed when different bonding tools are applied.
Please provide the manufacturers names of the used equipment (e.g. an ultrasonic bonder).
Author Response
We are grateful to the reviewer for the useful comments. This paper mainly focused on the bonding process of multi-layered foil, which is thinner than the metal sheets in our previous report, and it has multiple bonding interfaces. In addition, damage of the bonding foil and the bond formation in the multi-foils are discussed. According to these, we have made additions and revisions to descriptions in the sections of introduction and discussions. Conclusions have also been revised. The corrections are shown in red in the revised manuscript.
Response to specific comments
#1 What are the working frequencies of used K-type and C-type tools?
Response
In the experiment, we prepared two sonotrodes with K-type and C-type. Their resonant frequencies were adjusted to 21kHz +-0.1kHz by machining. The information has been added to “Experimental procedure”, page 3 line 87, in the revised paper.
#2 Please provide the manufacturers names of the used equipment (e.g. an ultrasonic bonder).
Response
The manufacturer’s name and model number of bonder have been added to page 3 line 85, in the revised paper. (USW1221G3X1, Ultrasonic engineering Co., Ltd)
Reviewer 2 Report
The paper deals with a methodology to bond a copper foil to a copper sheet by ultrasonic vibrations. The work describes the bonding method, observations of the assemblies obtained, and some strength tests to assess the validity of the proposed technique.
The paper is in general well written and well organized. Only minor revisions, mainly in the first part regarding the description of the method and of the used tools should be improved before publication:
Page 2, line 74: "and a 2.8-μm-thick Ni layer" it is not clear where the Ni layer is located
Page 2, line 74: it is said that "a 0.3-mm-thick copper sheet was used instead of the copper foil". Instead of what? Not clear
Page 2, line 85: "has a knurled pattern with edges". Which edges? What does it mean?
Page 3, line 86: "a radius of 200 mm" along which direction?
Page 3, line 87: what is the amplitude of the tool? In which direction?
Page 3, Figure 3: "in parallel to specimen" does not seem correct English and it is not even clear
Page 4, line 127: what does exactly mean "using a rectangle"? The expression should be better rephrased
For the rest, the paper is interesting and worth of publication.
Author Response
We are grateful to the reviewer for the useful comments and suggestions that have helped us to improve our paper. As indicated in the responses that follow, we have taken these comments into account in the revised version of our paper.
Comment #1. Page 2, line 74: "and a 2.8-μm-thick Ni layer" it is not clear where the Ni layer is located
Response: Ni layer with 2.8-µm-thick was plated on both side of the bonding surface and the lower surface in contact with the anvil. The information has been added to page 2, line77
Comment #2 Page 2, line 74: it is said that "a 0.3-mm-thick copper sheet was used instead of the copper foil". Instead of what? Not clear
Response: “The copper foil” has been revised to “the multilayered foil”. page 2, line 80
Comment #3 Page 2, line 85: "has a knurled pattern with edges". Which edges? What does it mean?
Response: The sentence has been revised to “The K-type tool shown in Fig. 2(b) has a knurled pattern. “ (“with edge” has been removed. )
Comment #4 Page 3, line 86: "a radius of 200 mm" along which direction?
Response: The radius is along a direction perpendicular to the vibration direction (direction y in the Fig.2) The information has been added to page 3, line 92
Comment #5 Page 3, line 87: what is the amplitude of the tool? In which direction?
Response: The direction of amplitude (vibration direction) measured has been added to page 3 line 93.
Comment #6 Page 3, Figure 3: "in parallel to specimen" does not seem correct English and it is not even clear
Response: "in parallel to specimen" has been removed. Instead, the direction x,y,z has been added in Fig.3
Comment #7 Page 4, line 127: what does exactly mean "using a rectangle"? The expression should be better rephrased
Response: The phrase has been changed to “marked by a red rectangle in Fig. 7(a)
Reviewer 3 Report
This document presents an investigation on the ultrasonic bonding of multi-layered copper foil to Ni-plated copper sheet.
Various comments will be useful for the authors:
1. Page 2. Line 73. There is no need to have a "-" between the value and the unit. It is better to write "0.2 mm thick" instead of "0.2-mm-thick". This happens here and at all text.
2. Page 5. Figure 6.
- The loads are good or not? I think it would be interesting if we know the required load for the joint.
- It is better to use different markers for C and K, because like that we understand the graph even on B/W printing.
Author Response
We are grateful to the reviewer for the useful comments and suggestions that have helped us to improve our paper. As indicated in the responses that follow, we have taken these comments into account in the revised version of our paper.
Comment #1 Page 2. Line 73. There is no need to have a "-" between the value and the unit. It is better to write "0.2 mm thick" instead of "0.2-mm-thick". This happens here and at all text.
Response: The errors have been revised as the reviewer pointed.
Comment #2.1 Page 5. Figure 6.
The loads are good or not? I think it would be interesting if we know the required load for the joint.
Response: We can conclude that the loads are sufficient (good) in all the joints just in the view of joint strength. However, it would be difficult to say the required load, quantitatively, because the fracture occasionally occurs at the foils even though the joint strength increased. Therefore we showed the joint strength comparing the result in C-type with that in K-type that is conventionally used.
Comment #2.2 It is better to use different markers for C and K, because like that we understand the graph even on B/W printing.
Response: As the reviewer suggested, we have revised Fig.6.
Reviewer 4 Report
The paper is well written and presents interesting topic. In my opinion it is worth for publication. The effect of the cylindrical surface tool on bondability was investigated thorough relative motion behaviors between the tool surface and the bonding materials, as well as on bond microstructure evolution. In my opinion, the paper meets all requirements for publication. I have only one minor comment to the paper:
Fig. 4. It is necessary to provide the main dimensions of tested specimen. Also, it is recommended to add the Force-displacement curves
Best regards,
Reviewers
Author Response
We are grateful to the reviewer for the useful comments and suggestions that have helped us to improve our paper. We have taken these comments into account in the revised version of our paper.
Comment #1 Fig. 4. It is necessary to provide the main dimensions of tested specimen. Also, it is recommended to add the Force-displacement curves
Response: The dimension of specimen used for the tensile test was the same as other bonding tests. Since it was difficult to know the dimension from the figure, we have revised Fig.4 as the reviewer suggested.
Round 2
Reviewer 1 Report
The authors have revised the manuscript according reviewer suggestions. Therefore, the paper should be accepted in the present form.